# Public Perception of the Range of Roles Played by Professional Pharmacists

**DOI:** 10.3390/ijerph16152787

**Published:** 2019-08-04

**Authors:** Anita Majchrowska, Renata Bogusz, Luiza Nowakowska, Jakub Pawlikowski, Włodzimierz Piątkowski, Michał Wiechetek

**Affiliations:** 1Independent Medical Sociology Unit, Chair of Humanities, Medical University of Lublin, 4-6 Staszica St., 20-081 Lublin, Poland; 2Department of Sociology of Health, Medicine and Family, Institute of Sociology, Maria Curie Skłodowska University, 4A Maria Curie Skłodowska Sq., 20-031 Lublin, Poland; 3Chair of Social Psychology and Psychology of Religion, Faculty of Psychology, The John Paul II Catholic University of Lublin, 14 Racławickie Av., 20-950 Lublin, Poland

**Keywords:** pharmaceutical care, consumer attitude, public perception

## Abstract

*Background*: Professional pharmacists should be directly involved in patient healthcare as members of therapeutic teams are not the only dispensers of medication. Public perceptions of the professional role of pharmacists is expressed through patients’ attitudes, trust, and expectations as health and illness consultants, or qualified retailers of medicines. This perception is influenced by numerous determinants, both health-related and social. *Objective*: This research intends to describe the range of social roles pharmacists play from the perspective of potential pharmacy customers/patients. *Methods*: The data presented in the article comes from cross-sectional survey-based research, undertaken in 2018, on a representative sample of 600 Polish adults. *Results*: Over-the-counter medication is purchased by almost all Polish adults, but they do not tend to ask for advice at pharmacies. Most respondents consider a pharmacist to be “a person qualified to sell medicines”, with some of the participants regarding pharmacists as “ordinary retailers”. A small number of respondents are interested in benefiting from pharmaceutical care, but the pharmacy is still perceived to be a point of purchase for medication. *Conclusions*: Respondents do not treat pharmacists as health advisors and reduce its role to that of dispensing medication. Sociodemographic variables have no significant effect on social perception of pharmacists.

## 1. Introduction

Ongoing transformations in the range of roles played by professional pharmacists have a global reach. According to the WHO, pharmacists should not only be dispensers of medication, but also members of a therapeutic team, actively tending to the healthcare needs of patients [1], who in turn benefit from the knowledge pharmacists have acquired in pharmacotherapy practice [2]. A special role is played by pharmacist employed in public pharmacies who deal directly with patients or customers. Increasingly, this kind of pharmacist is the only person in contact with patients suffering from less serious health problems, and they are generally the most important OTC (over-the-counter) advisor for such patients [3]. Selecting suitable OTC medication, monitoring purchases and usage, accessing pharmaceutical care and consultation, broadly conceived, should be everyday facets of the work undertaken by pharmacist employed at community pharmacies [4]. In many countries around the world, such as Great Britain, the United States, Australia, and New Zealand, pharmacists, write prescriptions for medication taken for minor ailments not requiring a medical diagnosis, review medication requirements, and perform personal pharmaceutical care. According to research conducted in these countries, patients accept that pharmacists are qualified to write out prescriptions, provide advice, offer health consultations and promote healthcare [5,6]. They are satisfied with the medication consultations they receive [7] and regard the pharmacist as a subject providing healthcare [8].

The benefits of pharmacists taking on more primary healthcare activities can be significant—both in terms of improving patient quality life and increasing financial benefits [9,10]. Unfortunately, the role of pharmacists employed at polish pharmacies is still limited to activities associated with the dispensing of medication, whether doctor-prescribed or OTC. Pharmaceutical care is conceived as a documented process, by which the pharmacist cooperates with a patient, doctor, or other medical professionals, to improve the effectiveness of pharmacotherapy, as well as the patient’s quality of life. This type of care is not currently widely available in Poland. A pilot program has been introduced in only a dozen or so Polish pharmacies, to encourage pharmacists to offer patients additional pharmaceutical services, including pharmacovigilance services, assessing home medical supplies, and taking pressure measurements. In the vast majority of Polish pharmacies, the role of pharmacist exclusively involves selling medicinal products, and it is this that determines both perceptions of this occupation and the nature of relations with patients/customers.

The perception of the range of roles played by professional pharmacists is influenced by numerous determinants, both health-related and social. In Poland and other Central European countries, an important role is also played by the socioeconomic transformations of the last century. In the period before the Second World War, a pharmacist running a private, usually family-managed, pharmacy served as a local authority and health advisor for many members of the immediate community. In the post-war communist administrations, the nationalization of pharmacies and mechanisms of the socialist economy (in 1951) changed the professional and social circumstances of pharmacists, who became employees of centrally-managed state pharmacies. Their role was limited to the ‘distribution’ and dispensing of medications prescribed by doctors. Losing their decisive role in pharmacy management and doctors’ dominance of the treatment process weakened pharmacists’ aspirations to describe themselves as a professional group, contributing to their occupation’s social devaluation. The re-privatization of pharmacies after Poland’s political transformation in the 1990s again changed the occupation of pharmacist’s conditions and range of responsibilities, allowing pharmacists more professional autonomy and opportunities to re-establish their standing among other medical occupations. A 2018 amendment to Polish pharmaceutical law, stipulating that a pharmacy can only be owned by a pharmacist [11], led to this occupation ‘coming full circle’: From chemist and owner of a pre-war pharmacy, to an employee of a state pharmacy in a socialist economy, and finally, a pharmacist-owner of a pharmacy in modern Poland.

Another important challenge in Poland is the organization of pharmaceutical care. Since 2008, this has been sanctioned by law as a duty of pharmacists themselves [12], yet in practice, most Polish pharmacies fail to provide it. Barriers hindering the implementation of pharmaceutical care within the Polish healthcare system include, organizational obstacles, such as pharmacists being afraid to take responsibility for patients’ health, not being convinced of their ability to provide effective care, and feeling anxious about effective cooperation with doctors [13,14]. The role and social standing of pharmacists is also influenced by patients’ relationships with pharmacists and their expectations surrounding the services they provide.

The objective of the conducted research was to describe how the public perceive pharmacists working at pharmacies, what expectations they have surrounding their skills, and the extent of their professional role. This is particularly interesting within the context of the political changes that have occurred in Central Europe, clearly evidenced by Poland’s example. A particular motivation for undertaking this research was the dearth of academic studies examining public perceptions of the occupation of pharmacist in Poland after the political transformations.

## 2. Material and Method

### 2.1. Study Participants

The research was carried out on 600 people over 18 years of age. The sample was random, and selection was calculated on the basis of them being representative of the Polish population in the following areas: Sex of respondents (100% compliance with calculations based on the Local Data Bank (LDB)), age of respondents (maximum deviation of 2% from calculations based on the LDB), the number of respondents in a given voivodship calculated on the basis of the population distribution throughout the country (100% compliance with calculations based on the LDB), place of residence (maximum deviation of 1% from calculations based on the LDB), level of education (maximum deviation of 3% from calculations based on the LDB).

The sample was primarily selected using random-route as a default method (employing the computer-assisted personal interview (CAPI) technique), while gaps in the metrical data (collected using the computer-assisted telephone interviewing [CATI] technique) were filled-in using a number generator to draw telephone numbers from a database of active numbers, issued by Polish landline and mobile operators. The maximum acceptable statistical error of measurement was 4% with a confidence interval of 95%.

### 2.2. Data Collection

The data comes from cross-sectional survey-based research carried out in 2018, using a mixed-mode survey technique, comprising 84% CAPI (the default technique) and 16% CATI (a secondary technique used to supplement the established metrical distributions). The questions included in the questionnaire would examine the respondents’ perceptions of the professional role of pharmacists. The study was a part of the wider research project on the perception of medical professions in Poland.

The three items that were assessed on a ranking scale examined the frequency with which respondents made over-the-counter (OTC) medication purchases and their willingness to use pharmaceutical care. The respondents were asked how often they bought OTC over-the-counter medication without consulting a doctor (possible responses: Several times a month, once a month, several times a year, once a year or less, or never). The next question related to consultation with pharmacists before buying OTC (“How often do you consult a pharmacist when buying OTC medication?”) could be responded by using the following scale: Always, often, sometimes, or never. The question about pharmaceutical care (“Would you use pharmaceutical care that entails regular meetings with a pharmacist, aimed at improving treatment quality and the safety of pharmacotherapy?”) could be responded using the following scale: “Certainly”, “I tend to”, “I tend not to”, “under no circumstances”.

The remaining statements were related to the perception of the pharmacist’s occupational and social role. In response to the question “who is a pharmacist working at a pharmacy is to you?”, respondents could indicate one answer from the following: “An ordinary retailer”, “a person qualified to sell medical products”, “a person able to substitute a doctor during the initial stage of treatment”, “a health consultant”, “a business owner trying to increase the pharmacy’s profitability”. Whereas, when answering the question: "What does a contemporary pharmacy primarily mean to you?", respondents could only choose no more than two of the following responses: “a profit-seeking enterprise”, “a place at which patients purchase medicines”, “a place where important decisions about patients’ health are taken”, “a place where doctor-prescribed medicines are prepared”, “a place where patients seek advice about treatment methods”, or “a drugstore/shop that sells other products/merchandise, as well as medicines”.

### 2.3. Statistical Analysis

The analysis was carried out using frequency measurements and a percentage of independent variables, as well as the respondent’s answers. The role of independent variables was assigned to sex, age, level of education, place of residence, self-assessment of health and material circumstances.

Due to the specificity of the analyzed variables, two non-parametric sets of statistics were used: Spearman’s rank correlation coefficient and the Mann–Whitney U test. A value of *p* < 0.05 was adopted as the level of significance. All data attained in the research was analyzed statistically using the IBM SPSS Statistics ver. 25 package (IBM, Armonk, NY, USA).

## 3. Results

There were 51% women and 49% men among the respondents. Respondents up to 29 years of age accounted for 17.7%, those up to 39 years accounted for 21%, those up to 49 years—17% and those up to 59 years—16.3%, while people aged 60 and over constituted 28%. While, 2.2% of respondents described their education as primary or gymnasial, 19.3% described it as basic vocational, 35.4%, as medium-level, 21.8% said it was a higher undergraduate, and 21.3% attained a higher master’s degree level. The vast majority of the respondents lived in cities, in most cases, large or medium-sized cities. The detailed characteristics of the studied group are presented in Table 1.

According to the research, nearly all Polish adults buy over-the-counter medication with different frequencies, with only 8.3% (*N* = 50) stating they “never buy over-the-counter medication”. A third of respondents do this at least once a month (34.9%; *N* = 209). A similar percentage state they make such purchases several times a year (34.5%; *N* = 207), while 22.3% do this occasionally (once a year or less). Younger people turn to over-the-counter medication much more frequently. A slightly higher prevalence of women buying OTC medicines is observable, but this is not a statistically significant difference (Table 2).

Polish patients are not inclined to seek advice at pharmacies. Only 2.3% (*N* = 14) of patients ‘always’ consulted a pharmacist when purchasing OTC medicine, 15% (*N* = 90) ‘often’ did this, and 56.3% (*N* = 341) ‘sometimes’ did. A quarter of respondents never sought advice. Women, people from small towns, and those who either have poorer health or are materially disadvantaged are much more likely to consult a pharmacist when buying OTC medication (cf. Table 2).

The vast majority of Poles think that pharmacists employed at pharmacies are people “qualified to sell medicines”, and for some potential pharmacy clients, they are “ordinary retailers”. Only a tenth of Poles tend to treat pharmacists as health advisors. Furthermore, the respondents primarily regarded pharmacies as places in which they bought medication as well as profit-seeking enterprises (Table 3). No correlation was found between perceptions of the occupation of pharmacist and the selected sociodemographic variables.

A small number of Poles are interested in taking advantage of pharmaceutical care, with only 16.5% (*N* = 99) of respondents stating they would decide to do this. Over half (58.7%; *N* = 352) would be “unlikely” to be interested in such care, while a fourth (24.8%; *N* = 149) would “certainly” not be interested. No correlation was noted between readiness to take advantage of pharmaceutical care and such variables as: Sex, age, education, place of residence, state of health, and material circumstances (Table 2).

## 4. Discussion

The conducted research indicates that Poles reduce the role of pharmacist to that of dispenser (or retailer) of medicines via a pharmacy. This predominant reductionist perception of the occupation limits community pharmacists’ range of responsibilities to dispensing medicine and sometimes provide information about medicinal products. The role of pharmacists is poorly understood, as is clear from the reluctance of most respondents to consult a pharmacist when purchasing medication or take advantage of pharmaceutical care.

It is extremely clear from the conducted representative research that Poles are not prepared to entrust a range of issues, related to their own health, to a pharmacist. The pharmacist is much more often perceived as a dispenser/retailer of medicinal products than a health and illness consultant. Interestingly, pharmacists perceive their role similarly. It is clear from the research conducted at Polish pharmacies that pharmacists do not perceive themselves as advisers of health and illness issues. For most pharmacists, administering pharmaceutical care and providing advice about self-medication issues are not what their occupation primarily entails. Instead, their main task is to dispense medicinal products, and provide information and advice related to medicines’ effects [15,16].

In other countries, where pharmacists’ new roles are developing more dynamically, some patients are not aware or need convincing of these roles, so fail to take the best advantage of the wide range of pharmaceutical services on offer [17,18,19]. Research undertaken in Malta shows that patients/clients mainly seek advice from community pharmacists about what to buy and how to use over-the-counter medicines, but if they need health advice, they tend to consult a doctor [20]. Nevertheless, pharmacy clients indicate that the most important growth areas in professional pharmacists’ role are liaising with doctors to support patients with chronic diseases and offering simple diagnostic test services (measuring blood pressure, sugar levels, etc.) [21].

In the UK, the community pharmacist’s role, as perceived by the public, ranged from 32% who saw pharmacists as primarily business people, to 26% who considered they were mainly concerned with health, and 42% who saw them as having a commitment to both health and business [22]. According to another study, members of the public rated the importance of pharmacists’ roles with regard to public health as relatively low, whilst activities linked to their traditional roles, such as advice on medication usage, side-effects, and disposal were rated the highest [23]. Research conducted by W. Gidman et al. indicates that patients still consider pharmacist’s main role just as medicines dispensation [24].

However, in some countries, public perceptions of pharmacists’ range of roles are broader. According to Australian research undertaken by E.C. Tan et al., patients in this country recognize the contribution pharmacists make to primary healthcare and express a high level of satisfaction with consultations conducted at pharmacies [25], although the community still regard a pharmacist’s primary role as dispensing medicines. On the other hand, research by K Hoti et al., indicates that Australian patients authorize pharmacists to write out prescriptions and provide advice [5], attesting to the high level of trust they place in pharmacists and their competencies. Patients from other countries also entrust pharmacists with a wide range of tasks involving patient care, health advice and health promotion, treating them as an integral part of a group of healthcare specialists [6,7,8,26,27,28,29]. Research conducted by Jaber at al. in Jordan indicates that, in developing countries as well, patients highly rate pharmacists’ pharmaceutical care competencies and tend to view their role as consultants rather than dispensers of medicines [30]. It is difficult to directly compare the studies cited above with Polish ones because they are based on different methodologies, and the patient/customer opinions of pharmacies in these countries are based on personal experiences, involving consultations with pharmacists administering pharmaceutical care not available to Polish patients.

Our results are limited by cross-sectional methods and it is impossible to establish trends and changes in social attitudes towards pharmacists. We cannot also compare our results with other research because studies of this kind have not yet been conducted in Poland. The comparative analysis is limited because of methodological differences (e.g., a shortage of quantitative studies on representative samples) in other countries make it impossible to directly compare our results with theirs. One of the limitations of the undertaken research was the use of non-standardised research tools, which prevents the comparison of the attained results with the data of other authors. Another problem is that the range of professional roles actually filled by Polish pharmacists, employed at pharmacies, is much narrower than in other countries, making direct comparison impossible.

## 5. Conclusions

Polish society tends to reduce the role of pharmacists to retailers/dispensers of medicines, and is not interested in pharmaceutical care or treating pharmacists as health consultants. Sociodemographic variables have no significant effect on this perception of the occupation of pharmacist. The research results indicate that the professional community of pharmacists is faced with the substantial challenge of changing their professional image in society: From retailers employed at a ‘drugstore’ to consultants responsible for patients’ health.

When designing future studies, it would be advisable to undertake a detailed analysis of the level of acceptance that both Poles and pharmacists express for each component of the expanded professional role of pharmacists employed at pharmacies. It would also appear to be extremely important to study the factors determining the narrow perception of the pharmacist’s role, identified in the current research. Many factors bound up with health provision in Polish society indicate a demand for the pharmacist’s role to be expanded, yet this demand is not expressed by patients or pharmacists. Evaluating the causes of this situation could help to improve patient-pharmacist relations, and result in both the expansion of the role played by pharmacists and improve their image within the community.

## Figures and Tables

**Table 1 ijerph-16-02787-t001:** Characteristics of the research group.

Feature	Demographic Variables	*N*	%
Gender:	Women	306	51.0%
	Men	294	49.0%
Age:	~29	106	17.7%
	30–39	126	21.0%
	40–49	102	17.0%
	50-59	98	16.3%
	60+	168	28.0%
Education:	Primary or gymnasial	13	2.2%
	Basic vocational	116	19.3%
	Secondary	212	35.3%
	Bachelor’s degree	131	21.8%
	Master’s degree	128	21.3%
Place of residence:	Village	33	5.5%
	Town with a population of up to 20,000	60	10.0%
	City with a population of between 20,000 and 100,000	228	38.0%
	City with a population of between 100,000 and 500,000	225	37.5%
	City with a population of 500,000 or more	54	9.0%
Self-assessment of health:	Very good	144	24.0%
	Good	372	62.0%
	Average	76	12.7%
	Poor	8	1.3%
Assessment of material circumstances:	Very good	68	11,3%
	Good	359	59.8%
	Average	166	27.7%
	Poor	7	1.2%

**Table 2 ijerph-16-02787-t002:** Sociodemographic variables and the behaviors of customers/patients coming into contact with a pharmacy.

Behaviors of Customers/Patients of Pharmacy	Spearman’s Rank Correlation Coefficient (rho)					Sex (Mann-Whitney U Test)			
						Median		Z	*p*
	Age ***	Education	Place of Residence	State of Health	Material Circumstances	Women	Men		
Purchase frequency of OTC medicines	−0.105 *	0.023	−0.035	0.063	0.065	3.12	3.01	1.751	0.08
Consulting a pharmacist about the purchase of OTC medicine	0.047	−0.080	−0.129 **	−0.089 *	−0.124 **	2.00	1.79	3.448	0.001
Willingness to take advantage of pharmaceutical care	−0.042	−0.020	−0.019	0.001	0.021	1.90	1.90	0.075	0.940

* *p* < 0.05; ** *p* <0.01; *** 10-year interval.

**Table 3 ijerph-16-02787-t003:** Respondents’ opinions of pharmacists and pharmacies.

Perceptions of the Pharmacist’s Role *	*N*	%	Perceptions of the Pharmacy **	*N*	%
A person qualified to sell medical products	427	71.2%	A place at which patients purchase medicines	506	84.3%
Ordinary retailer	85	14.2%	A profit-seeking enterprise	78	13.0%
A health consultant	63	10.5%	A drugstore/ shop that sells other products/merchandise as well as medicines	63	10.5%
A person able to substitute for a doctor during the initial stage of treatment	11	1.8%	A place at which patients seek advice about treatment methods	52	8.7%
A business owner trying to increase the pharmacy’s profitability	14	2.3%	A place where doctor-prescribed medicines are prepared	52	8.7%
			A place where important decisions about patients’ health are made	48	8.0%

* Respondents could choose only one answer; ** Respondents could choose no more than two answers.

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
