# Peer review of "Public Perception of the Range of Roles Played by Professional Pharmacists"

_ijerph, 2019, doi:10.3390/ijerph16152787_

Round 1

Reviewer 1 Report

While this is overall a fairly well written paper, the findings are not particularly surprising. Indeed, the assertion by the authors at the outset of the paper of the limited view that Poles have of pharmacists was borne out in the survey. Perhaps the point is that this prevailing notion has now been empirically demonstrated. Ironically, however, there seems to be congruence between what the public perceive the role of pharmacist to be, and the comfort level pharmacists have with that role. While the argument is made that there is room for an expanded role, it would appear that many pharmacists are reluctant to expand their scope of practice. What would have been much more interesting and helpful to the readership, I believe, is unpacking what is driving that reluctance and how it might be addressed.

Very limited information is provided about important aspects of the survey distribution and return, and this can be addressed by way of revision should you decide to publish.

I would suggest you have a statistician review for soundness if you decide to publish. My expertise is in qualitative methods, and while I have offered some comments regarding the design, do not purport to be an expert on the stats used for analysis.

Author Response

Dear Reviewer,

Thank you very much for the review of our manuscript. 

We are very much obliged for your thought-provoking comments on the article. The situation facing the profession of pharmacist in Poland is quite unique. On the one hand, there is a huge demand for the role played by pharmacists to be expanded (driven by increasing consumption of both doctor-prescribed and OTC medication, the prevalence of chronic diseases, an ageing population and dysfunctionality of the Polish healthcare system), yet on the other hand, a lack of interest in this form of pharmaceutical care can be observed, both among patients and among pharmacists who are actually employed at pharmacies. The lack of research interest in the profession of pharmacist (compared to the professions of doctor or nurse) means that there is little empirical basis for constructing assertions in this area. We hope that our interest in the profession of pharmacist and our research initiatives will contribute to the broadening of knowledge about this profession.

The comparison of the pharmacists’ professional role perception is presented in the Discussion section. Unfortunately, since it was necessary to use a bespoke, rather than standardised, research tool, it was not possible to undertake a precise comparison of the attained results with the data of other authors. The comparisons presented in the discussion do not, therefore, refer to identical (in the sense of being exactly the same) issues.

The research methodology will explained further in the corrected article.

We hope that the explanations we have given and the additions made to the text/article will prove to be satisfactory.

Kind regards,

Anita Majchrowska

corresponding author

Reviewer 2 Report

The manuscript needs important improvements, especially in the methodology section:

The use of MeSH descriptors is recommended as keywords.

The introduction gives a brief description of the role of pharmacists over the last decades. This contextualizes the phenomenon under study but requires a more in-depth analysis of the different national and international similar studies published to the proposed, on what is the perception of patients about the role of the pharmacist. In this way, it will be possible to justify the need for the study and describe the characteristics and responsibilities to be assumed.

The methodology presents some deficiencies that must be explained:

o   The type of study design should be indicated, which seems to be cross-sectional descriptive observational.

o   The scope of the study, the target population, the sample (specifying the population-based sample citation or intentionally), as well as the sampling done must be explained.

o   The inclusion and exclusion criteria must be specified, as well as how the subjects were recruited.

o   Information on the variables included in the study should be expanded.

o   The data collection instrument should be better explained and the scale of measurement. If it is validated or not. If it is not, how were the biases handled or resolved? If validated, include the validation data in a well-founded manner.

o   An explanation of the statistical methods used for the analysis, the main statistical tests used, must be included.

o   The use of nonparametric tests such as the Mann-Whitney U test must be justified

The table of sociodemographic characteristics should not be included in the methodology but should appear in the results.

In the limitations of the study, mention should be made of the weaknesses inherent in a graded study and the use of non-validated instruments.

The conclusions should provide recommendations for possible future studies on the phenomenon based on the results to guide potential researchers.

Author Response

Dear Reviewer,

Thank you very much for the review of our manuscript. We sincerely appreciate all valuable comments and suggestions, which helped us to improve the quality of the article. Our responses to the comment are described below in a point-to-point manner.

The introduction gives a brief description of the role of pharmacists over the last decades. This contextualizes the phenomenon under study but requires a more in-depth analysis of the different national and international similar studies published to the proposed, on what is the perception of patients about the role of the pharmacist. In this way, it will be possible to justify the need for the study and describe the characteristics and responsibilities to be assumed.

The comparison of the pharmacists’ professional role perception is presented in the Discussion section. Unfortunately, since it was necessary to use a bespoke, rather than standardised, research tool, it was not possible to undertake a precise comparison of the attained results with the data of other authors. The comparisons presented in the discussion do not, therefore, refer to identical (in the sense of being exactly the same) issues.

The methodology presents some deficiencies that must be explained:

1.     The type of study design should be indicated, which seems to be cross-sectional descriptive observational.

2.     The scope of the study, the target population, the sample (specifying the population-based sample citation or intentionally), as well as the sampling done must be explained.

3.     The inclusion and exclusion criteria must be specified, as well as how the subjects were recruited.

4.     Information on the variables included in the study should be expanded.

The research, which was nationwide and cross-sectional in nature, was carried out on a representative sample of 600 people. The sample was random and stratified, taking into account such variables as sex, age, level of education and place of residence. The sample was calculated on the basis of them being representative of the Polish population in the following areas: sex of respondents (100% compliance with calculations based on the Local Data Bank [LDB]), age of respondents (maximum deviation of 2% from calculations based on the LDB), the number of respondents in a given voivodship calculated on the basis of the population distribution throughout the country (100% compliance with calculations based on the LDB), place of residence (maximum deviation of 1% from calculations based on the LDB), level of education (maximum deviation of 3% from calculations based on the LDB). The detailed process of selecting the sample has been explained in the article.

5.     The data collection instrument should be better explained and the scale of measurement. If it is validated or not. If it is not, how were the biases handled or resolved? If validated, include the validation data in a well-founded manner.

The research addressed public opinion on medical professions using a bespoke questionnaire designed for the needs of the research by a team of sociologists. The questionnaire's suitability was discussed and approved by an expert panel and then verified practically during pilot studies. The lack of previous research of a similar nature on the profession of pharmacist made it impossible to use a standardised research tool.

6.     An explanation of the statistical methods used for the analysis, the main statistical tests used, must be included.

7.     The use of nonparametric tests such as the Mann-Whitney U test must be justified

 Due to the specificity of the analysed variables, two non-parametric sets of statistics were used:     Spearman's rank correlation coefficient and the Mann–Whitney U test. The questions used during the  research were assessed on a ranking scale. It was impossible to compute parameters for reliability because we assessed every variable (e.g. consulting a pharmacist about the purchase of OTC medicine)  using only one item. We agree that this may be illegible, which is why we changed averaged ranks to medians. We also made the presented results more readable by using non-parametric tests when possible.

8.     The table of sociodemographic characteristics should not be included in the methodology but should appear in the results.

Suggested change has been introduced to the manuscript (highlighted within the document).

9.     In the limitations of the study, mention should be made of the weaknesses inherent in a graded study and the use of non-validated instruments.

Suggested change has been introduced to the manuscript (highlighted within the document).

10.  The conclusions should provide recommendations for possible future studies on the phenomenon based on the results to guide potential researchers.

Suggested change has been introduced to the manuscript (highlighted within the document).

We hope that the explanations we have given and the additions made to the text/article will prove to be satisfactory.

Kind regards,

Anita Majchrowska

corresponding author

Round 2

Reviewer 2 Report

The authors have included the aspects considered in the review. The study methodology has been more adequately explained and the results are clearer. In addition, limitations and practical implications of the study have been included.

Only in the introduction, what was requested is that the authors include a more concrete description of the pharmacists' responsibilities in other settings according to published studies. We had not asked for a comparative study with the results, but a description of what roles and perceptions patients have in other countries and other studies.

Author Response

Dear Reviewer,

Thank you very much one more time for the review of our manuscript. We sincerely appreciate your valuable comment, which helped us to improve the quality of the article.

I hope the suggested changes have been made according to your wishes. The review pointed out the need to be more specific about the role played by pharmacists in Poland. The subject of the article and objective of the research were to describe the range of tasks performed by pharmacists employed at public pharmacies. Consequently, no reference was made to other roles played by pharmacists in, for example, hospital wards, hospital pharmacies or clinical research. We hope that the differences in how the occupation of pharmacist employed at a pharmacy is performed in Poland as opposed to other countries - essentially, pharmaceutical care and its various components are rarely offered in the former - will now be easier to perceive.  The lack of research interest in the profession of pharmacist (compared to the professions of doctor or nurse) means that there is little empirical basis for constructing assertions in this area. We hope that our interest in the profession of pharmacist and our research initiatives will contribute to the broadening of knowledge about this profession.

We hope that the explanations we have given and the additions made to the text  will prove to be satisfactory.

Kind regards,

Anita Majchrowska

corresponding author